# A Review on Soft Error Correcting Techniques of Aerospace-Grade Static RAM-Based Field-Programmable Gate Arrays

**DOI:** 10.3390/s24165356

**Published:** 2024-08-19

**Authors:** Weihang Wang, Xuewu Li, Lei Chen, Huabo Sun, Fan Zhang

**Affiliations:** 1Beijing Microelectronics Technology Institute, Beijing 100076, China; lixuewu1979@126.com (X.L.); sunhb@mxtronics.com (H.S.); zhangfan@mxtronics.com (F.Z.); 2China Academy of Aerospace Electronics Technology, Beijing 100094, China; chenleinpu@vip.126.com

**Keywords:** SRAM-based FPGA, sensitivity analysis, configuration scrubbing, memory scrubbing

## Abstract

Aerospace-grade SRAM-based field-programmable gate arrays (FPGAs) used in space applications are highly susceptible to single event effects, leading to soft errors in FPGAs. Additionally, as FPGAs scale up, the difficulty of correcting soft errors also increases. This paper proposes that performing soft error sensitivity analysis on FPGAs can help target the more sensitive areas for detection and correction, thereby improving the efficiency of soft error repair. Firstly, in accordance with the dual-layer architecture of SRAM-based FPGAs, methods for the soft error sensitivity analysis of FPGA application layer resources and configuration bitstreams are reviewed. Subsequently, based on the analysis results, it also covers corresponding application layer memory scrubbing and configuration scrubbing techniques. A prospective look at emerging soft error mitigation technologies is discussed at the end of this review, supporting the development of highly reliable aerospace-grade SRAM-based FPGAs.

## 1. Introduction

Field-programmable gate arrays (FPGAs) are extensively utilized in aerospace systems due to their reconfigurability, capability to perform complex functions, and their abundant logic, computation, and I/O resources [1]. These devices can dynamically adjust or upgrade the functions they perform depending on varying tasks and environments without the need for replacing the entire chip. Among the three common types of FPGAs available in the market (SRAM-, antifuse-, and Flash-based), SRAM-based FPGAs are particularly favored in aerospace missions for their ability to be configured almost indefinitely. However, a notable drawback of SRAM-based FPGAs is their susceptibility to errors or failures induced by radiation exposure [2,3]. Furthermore, as the scale of FPGA devices grows and the capacity of configuration bitstreams increases, the time required for reconfiguration lengthens, thereby reducing the efficiency of SEU (Single Event Upset) repairs. For example, CREME96 was run on the heavy-ion data to determine the SEU rates of Xilinx’s Kintex UltraScale+ FPGA in a geostationary orbit under solar minimum conditions, 100 mils of aluminum shielding [4]. The result shows that although the bit-wise event rates are relatively low (9.18 × 10^−12^, 2.08 × 10^−8^, and 5.29 × 10^−9^ upsets/bit/day in configuration RAM, Flip-Flops, and BlockRAM respectively), the device-wise event rates are remarkably high (1.64 × 10^−3^, 1.44 × 10^−2^, and 1.78 × 10^−1^ upsets/device/day in configuration RAM, Flip-Flops, and BlockRAM respectively). Therefore, more efficient, advanced fault correction techniques are needed.

To enhance the efficiency of repairing SEUs in FPGAs, it is crucial to specifically protect and repair the sensitive regions within the device. Therefore, soft error sensitivity analysis is often performed on FPGAs to estimate the likelihood of SEUs. The result of analysis can also be used to determine the level of impact of different parts on the circuit functionality. For SRAM-based FPGAs, which feature a dual-layer structure (i.e., application layer and configuration layer) as shown in Figure 1 [5], the soft error sensitivity analysis can be categorized into two types: analysis targeting the sensitivity of application layer resources and analysis targeting the sensitivity of the configuration bitstream.

More efficient FPGA soft error repairing methods could be derived by analyzing the results of soft error sensitivity. Researchers can prioritize the detection and repair of more sensitive areas in the application/configuration layer, enabling the faster identification and correction of soft errors induced by single event effects. Due to the high reliability demands in aerospace missions, it is particularly crucial to promptly detect, mitigate, and correct single event upsets (SEUs) induced errors in SRAM-based FPGAs. Commonly, FPGAs are protected by radiation hardening [6,7] or techniques like dual modular redundancy [8,9] and triple modular redundancy [10,11] to mitigate SEUs. However, these measures do not correct bit flips caused by radiation particles, which can accumulate over time and eventually lead to failures [12,13]. Therefore, it is necessary to promptly repair soft errors in FPGAs by locating and correcting the error data in the configuration random access memory (CRAM) or the storage resources in the FPGA application layer. This repair technique is known as “scrubbing”, which can be categorized into “configuration scrubbing” (also called “CRAM scrubbing”) and “memory scrubbing” depending on the location of the repair. Traditional configuration scrubbing methods were sufficient, but as FPGA scales continue to expand, these traditional methods struggle to meet the efficiency requirements for soft error repair. Advanced configuration scrubbing techniques improve upon traditional methods by designing nonlinear, high-level configuration scrubbing strategies based on FPGA configuration layer sensitivity analysis or by designing dynamic refresh scheduling mechanisms in multi-task scenarios. Research on memory scrubbing, based on the sensitivity analysis of application layer resources, targets on errors of resources in the FPGA application layer. By combining both scrubbing techniques, the system’s reliability could be further enhanced.

There have already been several review articles in this field. Nidhin, T. S., et al. introduced error correcting techniques in configuration memory and mitigation techniques of routing and logic resources in the application layer [14]. Similarly, Liu, Z., et al.’s review also involves techniques including TMR and scrubbing [15]. However, these reviews have no detailed introduction about more advanced scrubbing methods, like non-linear configuration scrubbing and multi-task scrubbing scheduling, which are contained in this paper. What is more, we point out that conducting soft error sensitivity analyzing can help design more effective repairing techniques. In this paper, we comprehensively review the soft error sensitivity analysis and repairing techniques of aerospace-grade SRAM-based FPGA.

The structure of this article is as follows. Section 2 of this paper summarizes the current SRAM-based FPGA soft error sensitivity analysis techniques. Section 3 reviews FPGA configuration scrubbing techniques, including basic methods and advanced nonlinear techniques. Section 4 reviews memory scrubbing techniques. Section 5 explores the prospects of more advanced soft error correcting techniques. Section 6 concludes this paper. In addition, for the sake of clarity, the structure of this paper is graphically illustrated in Figure 2.

## 2. SRAM-Based FPGA Soft Error Sensitivity Analysis Techniques

Soft error sensitivity analysis techniques for SRAM-based FPGAs can be categorized into two types: analysis targeting the resources of the application layer and analysis focusing on the configuration bitstream. The former assesses the SEU sensitivity of various resources within the FPGA application layer and examines whether these resources alter the normal functionality of the circuit when affected by SEUs. The latter investigates which data within the configuration bitstream would alter the circuits defined in the application layer or even lead to erroneous circuit outputs when flipped due to SEUs. This paper provides a comprehensive analysis and summary of the findings from both types of sensitivity analyses.

### 2.1. Sensitivity Analysis of Application Layer Resources in SRAM-Based FPGAs

The resource-oriented soft error sensitivity analysis for SRAM-based FPGAs focuses on different types of resources within the device’s application layer, aiming to analyze their failure rates and the extent of impact on the circuits when subjected to single event effects. According to FPGA testing guidelines [16], the resources in the FPGA application layer that require testing include Digital Flip-Flops (DFFs), BlockRAM (BRAM), clock networks, interconnect resources, and Input/Output Blocks (IOBs). The specific methods of analysis are primarily divided into two types: fault injection and analytical modeling.

#### 2.1.1. Fault Emulation-Based Soft Error Sensitivity Analysis

Fault emulation-based sensitivity analysis involves introducing errors into the FPGA application layer to emulate SEUs. This method assesses the sensitivity of resources by counting the number of soft errors in various components after the tests or by evaluating the extent of impact on the circuits when errors occur. Thus, it determines the criticality of the affected resources. The techniques for fault emulation are primarily categorized into two types: radiation experiments and hardware emulation.

Using radiation experiments for fault emulation involves exposing the FPGA under test to high-energy particles, inducing errors in the application layer resources of the FPGA. Fabero, J.C., et al. conducted neutron radiation tests with 14 MeV energy on Artix-7 FPGA devices and compared the number of soft errors in DFFs and BRAMs after the experiments (as illustrated in Figure 3) [17]. The results showed that DFFs exhibited only a few errors across ten trials, indicating low sensitivity to soft errors, and no errors were observed in BRAMs. Researchers speculated that this might be related to the inherent error detection mechanisms in BRAMs. Further studies conducted by Tsiligiannis, G., et al. on a similar FPGA device (XC7A25T) found that disabling error correcting code (ECC) significantly increased the probability of bit flips in BRAMs, supporting the aforementioned hypothesis [18]. This suggests that BRAMs are generally unaffected by SEUs due to their embedded ECC soft error detection mechanisms. For IOB resources, FZ Tazi et al. conducted proton radiation tests on Xilinx Virtex-5 and Virtex-7 series FPGAs [19] and discovered that SEEs (single event effects) induced additional delays in IOBs, with delays up to 6.2 ns and 3.8 ns in the respective devices. They also noted that although IOBs require fewer configuration bits than Configurable Logic Blocks (CLBs) and routing resources, the probability of soft errors occurring was comparable. Regarding interconnect resources, Darvishi, M., et al. focused on routing resources and the switch matrices (SMs) at the root nodes of the interconnect networks [20], describing how exposure to ionizing neutron radiation caused delay variations due to single particle disturbances. For clock networks, Guibbaud, N., and his team described a method for precisely measuring the cross section of various clock buffers in the clock network. His research conducts pulsed laser tests to achieve fully automated, highly reliable, and repeatable precise fault emulation [21].

Using hardware emulation for fault emulation involves emulating errors in data paths, control unit registers states, and internal signal states of the FPGA application layer. This allows for targeted sensitivity analysis of respective resources to soft errors. Mansour, W., introduced NETFI (NETlist Fault Injection), which allows for the modification of circuits at the netlist level to emulate errors, and these modifications can then be mapped to errors in the FPGA application layer after synthesis and implementation [22]. However, developing fault emulation tools from scratch is time-consuming and prone to bugs. To maximize the use of commercially available EDA tools, Ebrahimi, M., et al. utilized Altera’s debugging tools, the In-System Memory Content Editor (MCE) and In-System Sources and Probes (SAP) (Altera, San Jose, CA, USA), to emulate soft errors in FPGA Flip-Flops and other memory units [23]. This was achieved through highly automated fault emulation methods using TCL scripts. Similarly, Rhod, E., et al. modified extracted circuit netlists, emulating errors within the FPGA after implementation [24]. This method was tested on several benchmark circuits from ITC’99, demonstrating the versatility of the proposed approach.

In the sensitivity analysis of FPGA application layer resources for soft errors, conducting radiation experiments to emulate faults can most accurately simulate the radiation conditions encountered in aerospace applications. However, this method is costly, time-consuming, and often has destructive effects on the devices, with the location of fault emulation being uncontrollable. What is more, radiation experiments can affect both the application layer and configuration layer at the same time. Also, the errors induced in the configuration layer may cause further errors in the application layer, presenting certain difficulties in conducting precise sensitivity analysis. Hardware emulation offers strong repeatability and less damage to circuits but requires the design of complex dedicated hardware to perform the emulation tasks.

#### 2.1.2. Analytical Model-Based Soft Error Sensitivity Analysis

This type of analysis typically utilizes probabilistic or statistical theoretical models. Initially, models such as the Poisson distribution were employed to model the fault behavior of various circuit nodes (such as lookup tables, multiplexers, programmable interconnect points, etc.) under the influence of SEEs in order to estimate the soft error rate across different design nodes. Subsequently, mathematical tools like Markov chains were used to calculate the probability of error propagation, determining the likelihood that different resources will impact circuit functionality when a soft error occurs. Asadi, G., et al. calculated the error probabilities for all nodes (including MUX, PIP, LUT, clock networks, etc.) in circuits implemented in the FPGA application layer. By integrating the circuit netlist, they derived the error propagation probability in the data paths, thereby identifying the data paths more sensitive to SEUs [25]. Experiments on Xilinx XCV300 FPGA demonstrated that this estimation approach could achieve an average accuracy of 95%. Research conducted by Hogan, J. A., et al. also utilized Markov chains to analyze system reliability [26,27,28]. Mousavi, M., et al. proposed a universal computational model for SEU sensitivity [29]. This model divides the process from an SEU affecting the device to generating erroneous output into two stages: error generation in FPGA resources due to an SEU and the propagation of this error through subsequent circuits (as illustrated in Figure 4). $\varepsilon_{des}$ represents the probability of an SEU inducing an error in FPGA resources, while the design vulnerability factor (DVF) represents the probability of this error propagating through subsequent circuits and causing output errors. The final probability of an output error induced by an SEU is the product of these two parameters:$$DS=\varepsilon_{des}\times DVF,0\leq DS\leq 1$$

For the more advanced Xilinx UltraScale+ series of SRAM-based FPGAs, Sterpone. L., et al. utilized the VERIPlace tool to evaluate the sensitivity of various resources within the UltraScale+ FPGA application layer to SEUs [30]. This tool also simulated the internal propagation effects of SEUs within the FPGA designs to estimate the probability of errors in circuit outputs. Furthermore, the research compared the predictive outcomes of the VERIPlace tool with results from radiation experiments, validating the effectiveness of the proposed soft error sensitivity analysis methodology.

The analytical model-based analysis method reviewed above helps understand the propagation of soft errors in application layer circuits without causing damage to the circuits. However, it often provides the worst-case error probability, leading to inaccurate sensitivity analysis results. Additionally, due to the complexity of the models for soft error propagation in circuits, the analysis usually requires a significant amount of time. Currently, this analysis method is still in development.

### 2.2. Sensitivity Analysis of Configuration Bitstreams in SRAM-Based FPGAs

The analysis of soft error sensitivity for SRAM-based FPGA configuration bitstreams primarily investigates which bits in the FPGA configuration bitstreams can alter user-defined circuits when flipped and which bits can cause erroneous outputs in user-defined circuits upon flipping. The following sections review the classification methods for FPGA configuration bits from a criticality perspective and the methods for locating sensitive configuration bits within the bitstream.

#### 2.2.1. Configuration Bits Classification

The distinction in criticality of configuration bits in SRAM-based FPGAs stems primarily from three factors. First, the sensitivity to single-event effects varies across different types and locations of resources within the FPGA. Second, the circuits implemented in the FPGA utilize only a subset of the FPGA resources; therefore, only the configuration data corresponding to these resources will impact the circuit upon being flipped. Third, erroneous data in D-flip-flops or BRAMs of FPGA may be flushed out after several clock cycles, which can result in bit flips in their configuration data not leading to incorrect circuit outputs [31,32]. These factors contribute to varying impacts of different configuration data on the functions of circuits defined within the FPGA. Based on the degree of criticality, configuration bits can be categorized into: non-essential bits, which correspond to resources not used in the FPGA; essential bits, which configure resources that are utilized by the user-defined circuits; and critical bits, a subset of essential bits, whose flipping results in erroneous outputs of the circuit. The relationships among configuration data, non-essential bits, essential bits, and critical bits are illustrated in Figure 5.

Gear et al. further subdivided critical bits into primary critical bits and secondary critical bits based on the concept of soft error accumulation [33]. The definition of primary critical bits remains the same as critical bits described previously. Secondary critical bits, when flipped individually, do not cause erroneous outputs in the circuit. However, if another bit also flips due to a SEU, the accumulation of soft errors might lead to a circuit malfunction. This more detailed classification of configuration bits enables researchers to conduct more thorough analyses of soft error sensitivity and to design more precise and efficient soft error correcting strategies.

#### 2.2.2. Methods for Locating Essential and Critical Bits

Locating essential and critical bits within the configuration bitstream is crucial for designing effective configuration scrubbing techniques. The identification of essential bits typically involves studying the mapping relationships between the application layer and the configuration layer of SRAM-based FPGAs, with a knowledge of which resources are being used. For instance, Le Roux et al., after studying the mapping relationship between the LUTs and the bitstream in the Xilinx Virtex-5 series of FPGAs, reported that each LUT is configured by 64 configuration bits, and these bits are evenly distributed across four frames (as depicted in Figure 6) [34].

Xilinx’s “Essential Bit Technology” is capable of identifying essential bits within the configuration bitstream [35] and can work in conjunction with SEM IP to correct single or multiple bit flips in the CRAM. This method significantly reduces the failure rate (FIT, failure in time) of FPGA. Ding, Z., et al., focusing on the Virtex-5 series of SRAM-based FPGAs from Xilinx, established a mapping relationship table between programmable points, configuration options, configuration control bits, and configuration control bit values. They also developed a fully automated, distributed high-performance analysis architecture and offset formula theory, which are utilized to rapidly generate test cases, formulate mapping tables, and verify their correctness. The processing flow of bitstream decoding is shown in Figure 7 [36]. Research on the mapping relationships in Xilinx’s Virtex-7 series of FPGAs [37,38,39] has also been conducted, along with BitFREE, a parallel analysis method for mapping relationships between the application layer and configuration layer across most FPGA models, which greatly enhances the speed of analysis while ensuring universality [40].

To identify critical bits, the common method employed is fault injection. This involves flipping a specific bit in the configuration bitstream to emulate a Single Event Upset in the configuration memory. The bit is deemed critical if the outputs of the reference circuit and the tested circuit differ after injection. Research studies [41,42] targeting the Xilinx Virtex-5 and Virtex-7 series SRAM-based FPGAs used similar approaches for critical bit selection: initially, the mapping relationship is utilized to determine the configuration bits (i.e., essential bits) corresponding to used resources based on their coordinates in the application layer. A precise fault injection system is then developed to further filter out critical bits from these essential bits. Following the identification of critical bits by fault injection method, Gear, K. W., et al. leveraged this information to design more efficient FPGA refresh techniques [33,43,44]. Specifically, ref. [33] subdivides critical bits into primary and secondary critical bits, noting that comprehensive fault injection could be excessively time-consuming. Therefore, ref. [33] employs a bit-by-bit injection approach to select primary critical bits, while a probabilistic dual-bit injection strategy is designed for secondary critical bits, allowing for the calculation of the criticality of each configuration frame at a specified confidence level.

Another method for identifying critical bits involves utilizing the previously mentioned sensitivity analysis of application layer resources. This approach identifies the resources whose failure due to soft errors leads to incorrect circuit outputs. The corresponding configuration bits in the CRAM are then determined based on the application layer to configuration layer mapping relationships. However, this method is complex and often fails to yield precise results in the selection of critical bits, making it infrequently used in the localization of critical bits.

In the context of soft error sensitivity analysis targeting configuration bitstreams, there has been significant research into establishing the mapping relationship between the FPGA application layer and the configuration layer. This facilitates the extraction of essential bits information from the circuit’s netlist file. However, most studies focused on earlier FPGA devices, such as Xilinx’s Virtex-5 and Virtex-7 series, with limited attention on later FPGA series like Xilinx’s UltraScale and UltraScale+ FPGAs. Furthermore, while many studies employ fault injection to filter out critical bits from the essential bits, this approach, feasible for pre-Virtex-7 series FPGAs, becomes impractical as FPGA scales further increase due to the extensive time required for comprehensive bit-by-bit fault injection. Hence, exploring a faster and simpler method for identifying critical bits has become a key research focus.

This section provides a comprehensive review of SRAM-based FPGA sensitivity analysis techniques targeting application layer resources and configuration bitstreams. The strengths and weaknesses of specific methods involved are summarized in Table 1.

## 3. SRAM-Based FPGA Configuration Scrubbing

Configuration scrubbing is an effective error correction technique for the configuration memory within SRAM-based FPGAs. In some instances, the scrubbing process requires the interruption of the FPGA normal operations: a complete bitstream is injected into the configuration layer, applications are temporarily suspended, and then reinitialized. In other cases, the FPGA can perform readback operations in the background to read and verify the data in the configuration memory without disrupting the system’s normal operation. Also scrubbing techniques can benefit from the partial dynamic reconfiguration (DPR) feature of FPGAs, allowing for the scrubbing of the erroneous configuration frames while the circuits are running. DPR allows the rewriting of a subset of configuration frames, either while user design is suspended or while user design is operating. During the FPGA design stage, FPGA design tools (such as Xilinx Vivado) are used to partition the design into multiple reconfigurable modules (RMs). For each module, independent bitstream files are generated, including the bitstream for the static region and the bitstream for the reconfigurable region. At runtime, the partial bitstream is loaded into the target area through the reconfiguration controller, enabling dynamic partial reconfiguration. Current configuration scrubbing technologies can be categorized into basic, linear scrubbing techniques and more complex, advanced scrubbing techniques.

### 3.1. Basic Configuration Scrubbing Techniques

Basic configuration scrubbing techniques for SRAM-based FPGAs can be classified from various perspectives. Depending on the spatial relationship between the scrubber and the FPGA, scrubbing methods can be divided into external and internal scrubbing. Scrubbing can also be classified by triggering conditions into periodic scrubbing and corrective scrubbing. In terms of granularity, it can be categorized into device-level scrubbing, frame-based scrubbing, or mixed granularity scrubbing. By the end of this section, studies on redundant configuration scrubbing are also reviewed.

#### 3.1.1. External and Internal Scrubbing

External scrubbing mechanisms, typically implemented outside the FPGA that requires protection, are referred to as external scrubbers, as depicted in Figure 8a. These mechanisms communicate with the FPGA through interfaces such as JTAG (Joint Test Action Group) or SelectMap, monitoring and verifying the data within the configuration memory of FPGA.

Xilinx has introduced an external scrubbing mechanism in which the last two bytes of each frame in the configuration memory store a CRC value derived from the frame data. This CRC value is then compared with a newly computed frame CRC to verify whether the frame has been affected by SEU. The scrubber, as proposed in [45], acts as a bridge between the configuration memory and the FPGA being scrubbed. It employs a finite state machine to switch between states such as reading the configuration bitstream, configuring the FPGA, calculating CRC values, and comparing CRC values, thereby controlling the scrubbing process.

To address the limitation of only detecting SEUs as noted in study [45], Kumar, M., et al. implemented a 16-state finite state machine scrubber in another FPGA [46]. This scrubber reads the configuration register values and the configuration pins’ electrical signal levels of the target FPGA to detect the types of SEEs and to initiate corresponding scrubbing actions. This has enabled a more comprehensive solution for correcting soft errors in FPGAs.

In internal scrubbing, the scrubber is implemented within the target FPGA and utilizes the internal logic and routing resources of FPGA alongside the user-customized design. The scrubber accesses and manipulates the configuration data directly through the built-in Internal Configuration Access Port (ICAP) of FPGA. As depicted in Figure 8b, a portion of the configuration memory stores the configuration data for the scrubber, while another portion stores the configuration data for the user circuit. The internal scrubber performs scrubbing operations on the CRAM via the ICAP interface.

Heiner, J., et al. proposed a classical internal scrubber architecture [47], utilizing an 8-bit PicoBlaze processor [48] embedded within the configuration logic to control the scrubbing process. This setup leverages the flexibility of the processor to execute complex read-back and scrub operations. In comparison, Xilinx has developed a more generalized internal scrubbing architecture capable of correcting errors caused by single event effects or emulating single event upsets. The scrubber can be implemented using the internal programmable resources or an internal processor of FPGA, utilizing Xilinx’s FRAME_ECC primitive for error detection and correction [49].

Similarly, a scrubbing strategy presented in [50] also uses the FRAME_ECC primitive from Xilinx but consumes significantly fewer resources. Within the same FPGA model, the resources consumed by [50] are approximately 20% of those used in [47] and 39% of [49]. Xilinx’s SEM IP core provides advanced error detection, correction, and injection capabilities [51]. Also, Xilinx provide an SEU controller, a hardware module integrated in FPGAs, which is capable of error detection, correction, and reconfiguration [52]. To further accelerate the scrub rate while leveraging processor flexibility, Li, X., et al. combined internal scrubbing methods with the high-speed PCAP (Processor Configuration Access Port) interface in the Xilinx Zynq-7000 SoC series, significantly enhancing scrub speeds [53]. Additionally, Lu, Y., et al. tailored their approach for scenarios involving external processors cooperating with FPGAs [54]. Recognizing that processors in high-radiation environments often operate at frequencies significantly lower than RAM [55], they set the RAM in FPGA to operate at twice the frequency of the processor. The control module acts as a bridge between the scrubber and the BRAM for half of the time, and switches to scrubbing the FPGA during the other half. The control logic adjusts the sequence of external operations and scrubbing activities to prevent conflicts. This method enables both operations to appear simultaneous, ensuring that the scrubbing process introduces no additional delays.

In comparison, external scrubbing methods can protect the logic circuits controlling the scrubbing from SEU but will occupy additional on-board area and introduce extra latency. Internal scrubbing mechanisms have shorter latency and need no extra on-board area since the scrubbing control circuit is also located within the FPGA being scrubbed. However, the control circuit is also susceptible to SEU. In practice, a trade-off between performance and reliability is needed to select the most suitable scrubbing mechanism.

#### 3.1.2. Periodic Scrubbing and Corrective Scrubbing

Periodic scrubbing and corrective scrubbing represent two different methods of initiating scrub operations, differentiated by their triggering conditions. Periodic scrubbing, also known as blind scrubbing, involves rewriting all configuration data stored externally back into the configuration memory of FPGA at fixed time intervals. The advantages of this method include fast recovery, simple control, and minimal additional area usage. However, it requires a highly reliable and radiation-insensitive scrub controller and significant storage space to maintain standard configuration bitstreams. Determining the optimal scrubbing interval to maximize efficiency can also be challenging. To address these issues, Ahmed, A., et al. proposed a blind scrubbing strategy based on Magneto-resistive Random Access Memory (MRAM) [56]. In this approach, configuration data are stored in an MRAM matrix, allowing data selection by switching matrix rows and columns during the scrubbing process, thereby eliminating the need for additional scrub controllers and standard configuration bitstream, also enhancing the scrubbing speed.

Corrective scrubbing is more advanced and flexible compared to periodic scrubbing, as it is triggered only upon detection of SEEs. The specific detection mechanisms vary with different methods. For single event functional interrupts (SEFI), ref. [46] identifies various types of SEFIs by monitoring the values of internal FPGA configuration registers and the electrical levels of external pins, subsequently implementing repairing measures such as power cycling or reconfiguration. For SBU and multiple bit upsets (MBU), high-speed SRAM-type FPGAs typically employ single error correction and double error detection (SEC-DED) codes to verify each frame. The configuration data in each frame are encoded using an H matrix, and the resulting code is written into reserved parity bits within the configuration frame [57,58]. Single bit flips can be directly corrected through decoding based on the H matrix. For multiple bit flips, the H matrix generates eigenvalues that indicate the error locations. Corrective scrubbing uses parity codes and these eigenvalues to detect bit flips, allowing for either a full device scrub or targeted scrubbing of the affected configuration frames.

The corrective scrubbing method utilizing SEC-DED codes faces three primary challenges. First, SEC-DED codes cannot correct double bit flips, and they may misjudge scenarios involving more than two flips. Second, this approach requires the reservation of parity bits within the configuration frames. Third, the encoding process of SEC-DED codes introduces significant delays, which can impact FPGA performance. To address these shortcomings, research has explored alternative parity codes for corrective scrubbing of FPGAs. For example, Rahul, K., et al. proposed an ECC that does not require additional parity bits and is capable of detecting and correcting adjacent 2-bit errors, as well as detecting adjacent 3-bit errors [59]. Additionally, they introduced an ECC that does require extra parity bits but can detect and correct adjacent 2 and 3-bit errors. To tackle the issue of encoding delays, Sen, P., et al. improved upon the decimal matrix code (DMC) [60], significantly reducing the time required for encoding while requiring fewer additional bits of information and handling a broader range of MBU errors.

#### 3.1.3. Device-Level Scrubbing, Frame-Based Scrubbing, and Mixed Granularity Scrubbing

From the perspective of scrubbing granularity, refreshing the entire device and refreshing by frame represent two contrasting methods. The former involves rewriting all the gold configuration data, stored in externally radiation-hardened memory, back into the configuration memory of FPGA upon detection of SEU or MBU. This approach is relatively straightforward but introduces additional and unnecessary delays and power consumption. It also disrupts the normal functions of the FPGA. In contrast, frame-based scrubbing offers more flexibility but has a comparatively complex control mechanism. The scrubber determines the occurrence of bit flips by reading the ECCs of each frame. If a flip is detected, the correct configuration data is rewritten into the affected frame. This process leverages the dynamic reconfiguration capabilities of FPGA to perform the correction without interrupting the FPGA’s normal operations.

The granularity of scrubbing is not fixed. Research presented in [61] introduced a hybrid granularity scrubbing method. This method initiates a fine-grained scrubbing mechanism when soft errors are detected in the CRAM, specifically refreshing the affected configuration frames. The coarse-grained scrubbing mechanism, on the other hand, refreshes configuration bits corresponding to a particular circuit module or the entire configuration bitstream. This approach is capable of detecting and correcting errors that cannot be identified or repaired through fine-grained repairs alone. The hybrid granularity scrubbing method addresses several issues: it reduces the time-consuming process of locating erroneous frames and fixes certain types of errors that fine-grained scrubbing cannot. Simultaneously, it resolves the low efficiency of coarse-grained scrubbing repairs. This method is particularly advantageous in large-scale FPGAs with millions or even hundreds of millions of gates.

#### 3.1.4. Redundant Configuration Scrubbing

Most of the configuration scrubbing techniques need a “golden bitstream” to be stored in a radiation-hardened memory. However, this cannot 100% prevent the golden bitstream being affected by SEU. For techniques that rely on error-correcting codes and thus do not require a golden bitstream, MBUs remain a challenging issue. To address this issue, scrubbing techniques based on “redundant configuration” were developed. Redundant configuration can be further categorized into “bit-level”, “frame-level”, and “device-level” redundancy.

Steiner, G. C., et al. introduced a design implementation flow aimed at generating redundant configuration at the bit-level in a programmable logic device (PLD) including FPGAs [62]. The configuration data are stored in the first portion of the configuration memory cells of the PLD, defining the logic function. Then, the unprogrammed configuration memory cells are identified. By programming the unprogrammed cells with the same configuration data as their corresponding programmed cells, this method implements bit-level redundancy. Homologous bits are the inputs of a majority voter, and the voter’s output determines the actual behavior of the PLD. This technique, disclosed in a patent [62], masks errors but is unable to detect or correct them.

Tonfat, J., et al. introduced a self-correcting method named “frame-level redundancy scrubbing (FLR-scrubbing)” [63]. Based on a coarse TMR design in FPGA, FLR-scrubbing involves replicating the target configuration frames cluster three times and storing them in three TMR domains in the CRAM of a single FPGA. The scrubber starts from the first frame of each frame cluster, and then executes majority vote to detect and correct the faulty configuration bits. The energy consumption is six times lower compared to the blind scrubbing technique. Similar frame-level redundancy scrubbing techniques are also proposed in [64,65]. Furthermore, a detailed solution for generating frame-level configuration redundancy is proposed in [66].

The device-level redundant configuration scrubbing involves implementing the same design in multiple identical FPGAs. Giordano, R., et al. introduced an error-correcting technique implemented in a system with six identical FPGAs [67]. By reading back and voting frames of the same address in the CRAMs, this method can realize rapid error correction and high reliability. Similarly, Herrera-Alzu, I., et al. and Alfke, P. H., et al. also proposed device-level redundancy methods, utilizing TMR principles [68,69].

### 3.2. Advanced Configuration Scrubbing Techniques

Basic scrubbing techniques typically employ fixed scrubbing sequences and frequencies, using static scheduling algorithms when faced with multiple scrubbing requests. This approach limits the exploration space for mean time to detect (MTTD) or mean time to repair (MTTR) and makes it challenging to optimally schedule scrubs in response to dynamically changing scrubbing requests, thereby compromising system reliability.

This section introduces advanced scrubbing techniques including nonlinear configuration scrubbing and multitasking scrub scheduling. Compared to basic scrubbing methods, these advanced techniques significantly enhance scrubbing efficiency and system reliability.

#### 3.2.1. Nonlinear Configuration Scrubbing

The analysis of soft error sensitivity with respect to configuration bitstreams shows that different configuration data vary in their impact on circuits implemented in FPGAs. Therefore, during the scrubbing process, it is advisable to prioritize frames containing essential or critical bits. These frames should be inspected and scrubbed first or subjected to higher detection frequencies, leading to the concept of “nonlinear scrubbing”. Nonlinear scrubbing breaks away from the traditional approach of basic scrubbing techniques, which start from the first frame and incrementally check each subsequent frame address. By altering the starting frame address and employing jumping read-back verification, nonlinear scrubbing explores further enhancements in scrubbing efficiency.

Nazar, G. L., et al. proposed the “Shifted Scrubbing” [43] method, which utilizes the non-uniform distribution of critical bits within the configuration bitstream—namely, the differing number of critical bits in each frame. This characteristic allows each frame’s proportion of critical bits in the total number of critical bits within the CRAM to represent the probability of a critical bit flip (a flip affecting circuit output) occurring in that frame. Based on this, the MTTR can be calculated under different starting frame addresses, from which the optimal starting frame can be derived. Figure 9 from [43] illustrates the distribution of critical bits across different frames in the benchmark circuit misex3. However, this study only altered the starting position of the scrubbing, and the read-back verification during the scrubbing process still followed a linear increase in frame addresses.

Building on the work of [43], Mousavi, M., et al. from Eindhoven University of Technology proposed the “Scatter Scrubbing” method [44]. This approach involves three steps. Firstly, identifying critical bits through fault injection and calculating frame criticality. Secondly, using this criticality data to develop a heuristic algorithm for the optimal grouping of configuration frames. And finally, applying nonlinear discrete optimization techniques to mathematically derive the MTTR formula and determine the optimal scrubbing sequence. Experimental results indicate that scatter scrubbing can reduce MTTR by an average of 40% and 25% compared to traditional read-back scrubbing and shifted scrubbing, respectively. The advantage of this method lies in its substantial exploration of enhanced scrubbing efficiency. However, its drawback includes additional delays introduced during frame address jumping (in the case of Xilinx 7 series FPGA, this delay is approximately 1.5 times the read-write time of a frame, because the FPGA needs extra time to read and process the new data in the frame address register). Also, the extent of MTTR reduction can vary significantly depending on the specific circuit, which somewhat limits the performance of the scrubbing algorithm.

Rongsheng Zhang et al. have noted that the accumulation of non-critical bit errors in the configuration memory of complex FPGA circuits may lead to unpredictable issues, prompting them to propose a combination of essential frame scrubbing and full device scrubbing. By scheduling N essential frame scrubs before a full device scrub, they further enhanced the reliability of the FPGA [70].

Building on the research in Refs. [43,44,70], Kyle W. Gear further subdivided critical bits into primary critical bits (PCB) and secondary critical bits (SCB) [33] (as shown in Figure 10). They quantified the “frame criticality” of each configuration frame based on the number of primary and secondary critical bits it contains, to weight its importance during the scrubbing process. The scrubbing sequence is determined based on the calculated frame criticality. The method for calculating frame criticality is illustrated in Equations (1) to (3). Equation (1) represents the count of all SCB pairs in a particular frame. Equation (2) calculates the criticality of the frame, i.e., the proportion of this frame relative to all frames. Equation (3) standardizes the criticality calculated in Equation (2).
(1)$$\mathrm{FrameSCBs}=\sum_{j=1}^{n}{\mathrm{SCB}}_{\mathrm{Pairs}}\left(i\right)$$
(2)$$\mathrm{Criticality}\left(f\right)=\frac{\mathrm{FrameSCBs}\left(f\right)}{\sum_{x=1}^{m}\mathrm{FrameSCBs}\left(f\right)}$$
(3)$${\mathrm{Crit}}_{\mathrm{Normalised}}\left(f\right)=\frac{\mathrm{FrameSCBs}\left(f\right)}{{\mathrm{FrameSCBs}}_{\mathrm{Min}}}$$

Similarly, He, G., from Shanghai Jiao Tong University in China categorized configuration frames into unused frames, essential frames, priority essential frames, and critical frames (as shown in Figure 11) [71]. In this study, configuration frames with a MTTM (mean time to manifest) shorter than MTTD are designated as priority essential frames, among which those with the smallest MTTM values are marked as critical frames. Frames that configure user-defined circuits and have an MTTM greater than the MTTD are categorized as essential frames. Frames not involved in circuit configuration are designated as unused frames. By implementing jumping scrubbing to achieve varying scrub detection frequencies, the scrubbing efficiency of FPGAs can be enhanced. The specific order of scrubbing can be determined through methods such as brute-force enumeration, dynamic programming algorithms, or sorting optimization algorithms.

Similar to [71], Mousavi, M., et al. also described the structure of CRAM configuration data in FPGAs from the perspective of SEU sensitivity [72] and further distinguish highly critical bits within the critical bits. They proposed more advanced shifted scrubbing and scatter scrubbing with consideration of precision levels. Experimental results showed that compared to basic read-back scrubbing methods, the MTTR was reduced by 24% to 46.5%. Additionally, for circuits with non-uniform structures such as FFT circuits (where the addresses of the configuration frames used in the CRAM are relatively concentrated), these advanced scrubbing methods achieved even smaller MTTR, confirming their effectiveness.

The nonlinear configuration scrubbing methods reviewed above are summarized in Table 2.

#### 3.2.2. Multitasking Scrub Scheduling

Most research on scrubbing techniques has been conducted under the assumption of a single hardware task (i.e., the user-defined circuit in FPGA), with a fixed scrubbing cycle for FPGAs. However, in practice, multiple tasks can be run on a single FPGA, dividing the application layer of the FPGA into several regions, each dedicated to implementing the circuit for a corresponding task. Scrubbing measures for these hardware tasks involve only detecting and repairing the configuration frames specific to each partition (as shown in Figure 12). In systems operating in real-time, multiple tasks may occur periodically or randomly, and they can vary in criticality. Frequently scrubbing a task that is not active results in a waste of resources. Also, applying the same scrubbing frequency to tasks of high and low criticality decreases system reliability. In this scenario, it is necessary to design an optimal scheduling method for multiple scrubbing tasks to achieve the highest system reliability.

Santos, R., from the National University of Singapore pioneered the study of multi-task scrubbing scheduling in FPGAs, linking the scrubbing process with the significance and timing of hardware tasks within the FPGA [73]. He developed a heuristic scrubbing schedule design algorithm based on task criticality and execution time. This method first calculates the minimal scrubbing period using integer linear programming based on the criticality of each task. It then employs the earliest deadline first (EDF) algorithm to schedule the scrubbing tasks, arranging for the end of the scrubbing tasks to be as close as possible to the commencement of the corresponding hardware tasks to minimize the likelihood of SEU impacts on the system. Experiments conducted in a multi-hardware task scenario compared this scheduling algorithm with traditional blind scrubbing and selective scrubbing approaches. The results indicated that compared to the blind scrubbing method, system reliability increased by 70–79%, and relative to selective scrubbing, reliability improved by 25–34%.

Although [73] effectively linked scrubbing tasks with hardware tasks to enhance system reliability in multi-task scenarios, the scrubbing scheduling algorithm employed was static, meaning it was calculated offline and could not adapt to the addition of new hardware tasks. Moreover, it required substantial space to store the pre-computed scrubbing schedule. To address these issues, Santos, R., further developed a dynamic adaptive scrubbing scheduling algorithm for multi-task scenarios [74], based on a time-window approach. This method divides time into a series of consecutive, fixed-length segments (i.e., “windows”), during which the system plans and executes scrubbing operations based on the dynamic changes of system status and hardware tasks. Experimental results demonstrated that this method could adapt to scenarios where new hardware tasks are added to the task set. System reliability was comparable to that of [73]. However, as it only required storage for the upcoming few windows rather than for the entire duration of FPGA operation, the required storage space (averaging 1.3 KBytes) was significantly reduced compared to [73] (averaging 21.3 KBytes).

Building on [73,74], Santos, R., et al. expanded the scope to include DSP applications and high-data-throughput tasks such as video processing and data acquisition [75]. Additional challenges in [73,74] include the limited dynamic reconfiguration capabilities of FPGAs, which reduce system reliability under the scenarios of the frequent parallel execution of hardware tasks. And for tasks of low criticality, the extended interval between two scrubbing operations makes them more susceptible to SEUs. To address these issues, Li, R., et al. from ShanghaiTech University employed dynamic voltage and frequency scaling (DVFS) to dynamically schedule hardware tasks implemented in FPGAs, aiming to reduce conflicts with scrubbing tasks [76]. Compared to [73], this scheduling algorithm improved system reliability by 15.51%.

In subsequent research, Li, R., pointed out that existing multi-task scrubbing scheduling algorithms either struggle to optimally manage conflicts among multiple scrubbing tasks in scenarios with multiple burst tasks or allocate low scrubbing frequencies to low-criticality user tasks, thereby reducing system reliability. To address these challenges, Li, R., proposed a negotiation-driven scrubbing scheduling algorithm [77], which temporarily tolerates conflicts between scrubbing tasks and resolves them iteratively, thereby enhancing the flexibility of the scheduling process and the reliability of the system. A logical probability model was developed to prevent potential scheduling starvation issues. Furthermore, utilizing the hardware characteristics of FPGAs, a DVFS-based multi-ICAP allocation algorithm was designed to maximize system reliability. Experimental results on a Xilinx Virtex-6 FPGA demonstrated that, compared to previous scrubbing scheduling methods, this approach could increase system reliability by up to 31.46%.

Table 3 provides a summary of the research for multi-task scrubbing scheduling techniques.

## 4. Memory Scrubbing Techniques

When soft errors occur in FPGA application layer storage units such as BRAMs and LUTs due to SEEs, the most direct method for error correction is configuration scrubbing. However, this method leads to interruptions in system functionality and is rarely used for repairing soft errors in the application layer storage. An alternative approach involves performing configuration scrubbing on the FPGA while employing redundancy-based soft error mitigation methods for user data within the application layer [78,79]. However, this can lead to the accumulation of errors and result in unpredictable outcomes. Consequently, there is a need for a method that specifically refreshes the user data stored in the FPGA application layer, termed “memory scrubbing.”

Rollins. N., et al. noted that memory scrubbing and configuration scrubbing are two fundamentally different soft error correcting techniques. Due to the dynamic nature of user data in the application layers of FPGAs, it is impractical to use golden data to detect errors in BRAMs or LUTs [80]. Further, Rollins, N., has proposed specific memory scrubbing techniques for different storage units within the FPGA application layer, including BRAM, LUTRAM, and Shift Register LUT (SRL). For BRAM, which has a larger storage capacity and is susceptible to errors at multiple locations due to SEUs, it is necessary to perform regular scrubbing. The method described in [80] utilizes triple modular redundancy or ECC in BRAM supported by FPGA vendors to detect errors and scrubs the data by writing through one of the BRAM ports. In contrast, smaller units like LUTRAM and SRL are scrubbed only during use. It is worth noting that accessing BRAM configuration conflicts with the operation of logic in the fabric accessing the pertaining memory blocks. Therefore, care should be taken to resolve these conflicts for successful operation of the design and of the configuration scrubber. 

The same author also investigates the reliability of the LEON3 (Atmel, San Jose, CA, USA) soft-core processor implemented in SRAM-based FPGAs, proposing a hybrid soft error mitigation strategy that combines parity, redundancy, checkpointing, and memory scrubbing [81]. Similarly, Wirthlin, M. J., et al. suggests a method of using triple modular redundancy combined with scrubbing to mitigate soft errors in the FPGA-based LEON3 processor [82]. This approach involves alternating read and write operations at each address of the BRAM to detect and repair potential soft errors, as illustrated in Figure 13. Both fault injection tests and irradiation experiments confirm that the proposed memory scrubbing technique enhances system reliability.

For BRAM memory scrubbing, studies [80,81,82] all rely on utilizing one of the BRAM’s ports to write correct data back to the corresponding addresses. This method is unfeasible when the BRAM is configured in single-port mode or when both ports are utilized in dual-port mode. Gomez-Cornejo, J., et al. have introduced an innovative approach for modifying BRAM data, which allows the contents of the BRAM to be extracted, loaded, or compared directly from the bitstream without utilizing the BRAM ports [83]. This technique, grounded in a thorough understanding of the configuration bitstream’s organizational structure, analyzes the addresses corresponding to BRAM data within the bitstream. By directly modifying the data at these specific locations within the configuration bitstream, BRAM data can be altered using dynamic reconfiguration techniques.

Based on [83], Gomez-Cornejo, J., targeted the Xilinx Zynq SoC, employing the SoC’s internal hard-core processor to analyze the bitstream and control the BRAM memory scrubbing process [84]. For distributed storage units within the FPGA application layer, the research employs the GSR signal within the STARTUPE2 primitive to reload initial values and repair potential soft errors. Experimental results indicate that this method reduces the BRAM scrubbing time to approximately 22.3% of that required by traditional BRAM scrubbing methods. Similarly, the scrubbing of distributed storage units is also exceptionally fast, approximately 0.4 μs, achieving trhe rapid repair of soft errors in the FPGA application layer.

## 5. Challenges and Future Directions

To date, extensive research has been conducted on the analysis and correcting techniques for soft errors in SRAM-based FPGAs. However, due to the complexity of FPGA devices, it is challenging for a sensitivity analysis to cover all resources comprehensively, and the accuracy of such analyses can be uncertain. Furthermore, as FPGA technology continues to evolve and application layers expand to billions of gates, the corresponding configuration bitstream size increases, thereby elevating the difficulty of detecting and correcting soft errors.

In summary, the journey toward fully understanding and mitigating soft errors in FPGAs remains long and complex. Based on the current research foundation, future trends in this field can be summarized as follows:(1)Develop more detailed and accurate models for soft error analysis in FPGA application layers. Current technologies struggle to accurately emulate the operational processes and states of circuits during functioning. Consequently, it is difficult to precisely determine which resources have a more significant impact on circuit performance when soft errors occur. Moreover, conducting more refined emulations requires substantial computational resources and time. Therefore, developing more accurate models for soft error analysis and enhancing the speed of such analyses represent one of the future directions for research in this field.(2)For memory scrubbing techniques, expand coverage to a broader range of storage units. Compared to configuration scrubbing, research on memory scrubbing is relatively scarce and has mostly focused on major storage units in the application layer, such as BRAM and LUTs. However, there has been limited research on the correction of soft errors in crucial but less data-intensive storage resources, such as control registers, which are critical for operation. Developing methods to detect and promptly correct soft errors in these key registers may become a focal point in future research on FPGA soft error mitigation techniques.(3)Incorporate artificial intelligence into the mitigation, detection, and repair of soft errors in FPGAs. The task of optimizing the layout of user circuits within FPGAs to minimize sensitivity to SEE is a heuristic problem with a vast variable space, making it challenging to find optimal solutions. Artificial intelligence is well-suited for solving such high-dimensional data space issues [85]. AI can be utilized to analyze designs within FPGAs, identify vulnerabilities in the circuits, and suggest appropriate improvements.

## 6. Conclusions

As the design technology of SRAM-based FPGAs continues to mature, the increasing scale and complexity of FPGAs have increased demands for soft error correction techniques in aerospace applications. This paper focuses on the soft error correction techniques for FPGAs, analyzing and summarizing existing research from two perspectives: FPGA soft error sensitivity analysis and correction techniques. It demonstrated that the results of FPGA soft error sensitivity analysis contribute to the design of more efficient correction methods. Finally, based on the current technical challenges, this paper forecasts the development trends of more efficient and intelligent soft error correction techniques, aiming to provide valuable insights for future research in this field.

## Figures and Tables

**Figure 1 sensors-24-05356-f001:**
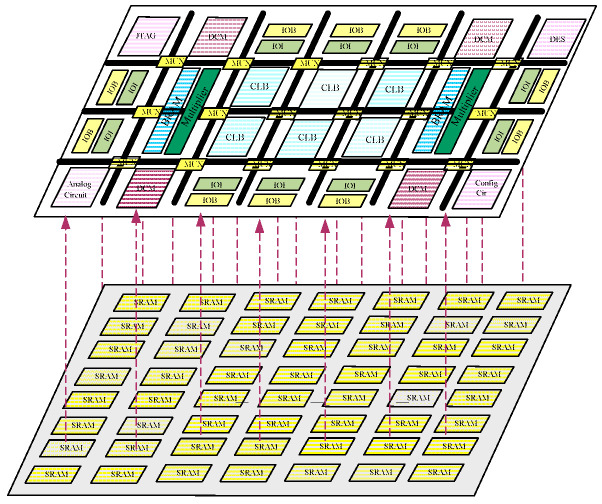
Dual-layer structure of SRAM-based FPGA.

**Figure 2 sensors-24-05356-f002:**
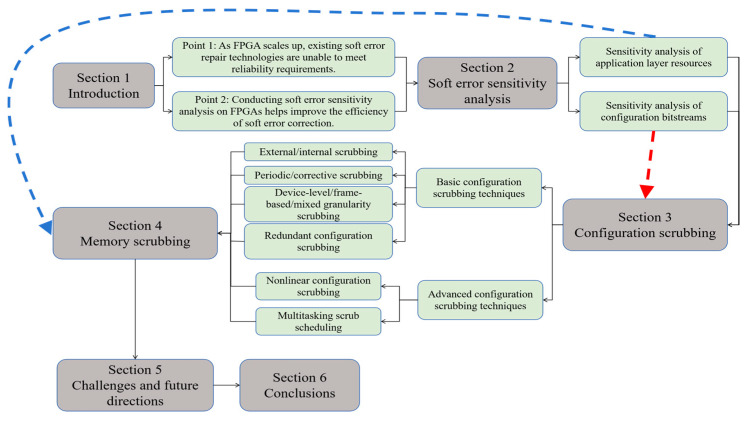
Structure of the paper.

**Figure 3 sensors-24-05356-f003:**
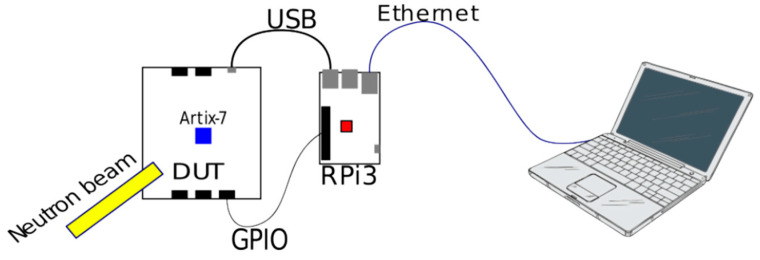
Experimental setup in [17].

**Figure 4 sensors-24-05356-f004:**
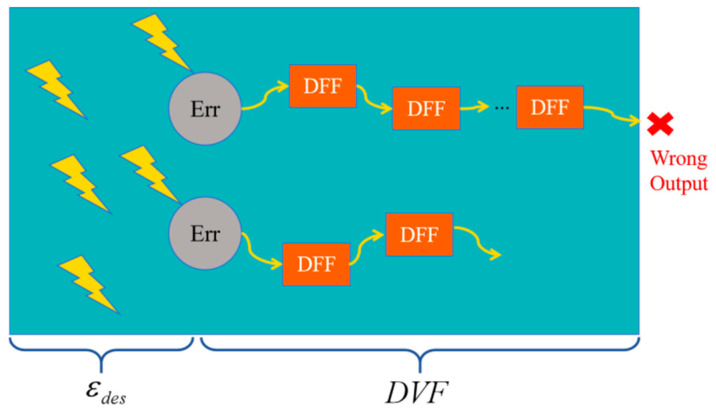
SEU sensitivity calculation model.

**Figure 5 sensors-24-05356-f005:**
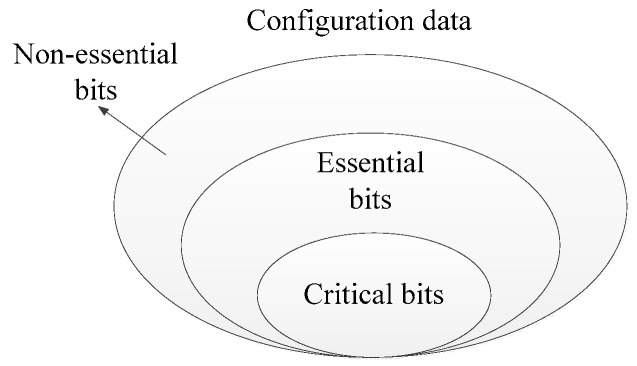
Relationships among configuration data, non-essential bits, essential bits, and critical bits.

**Figure 6 sensors-24-05356-f006:**
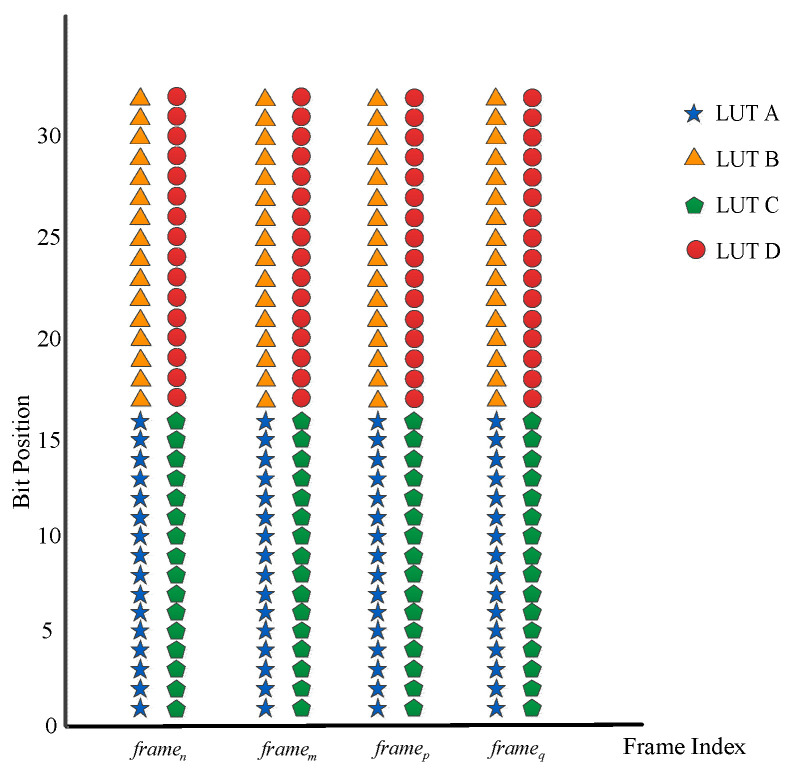
LUT—configuration bits mapping.

**Figure 7 sensors-24-05356-f007:**
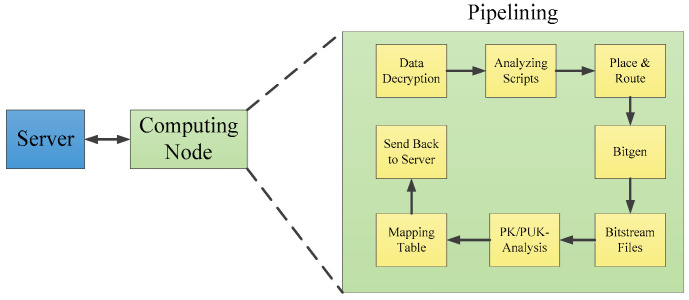
Bitstream decoding process.

**Figure 8 sensors-24-05356-f008:**
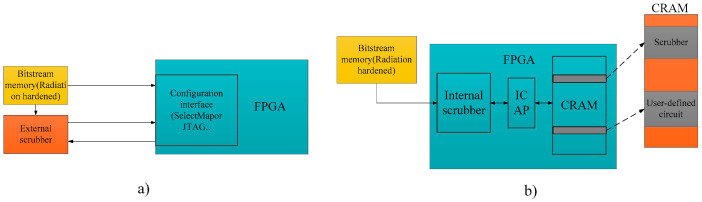
External scrubber and internal scrubber.

**Figure 9 sensors-24-05356-f009:**
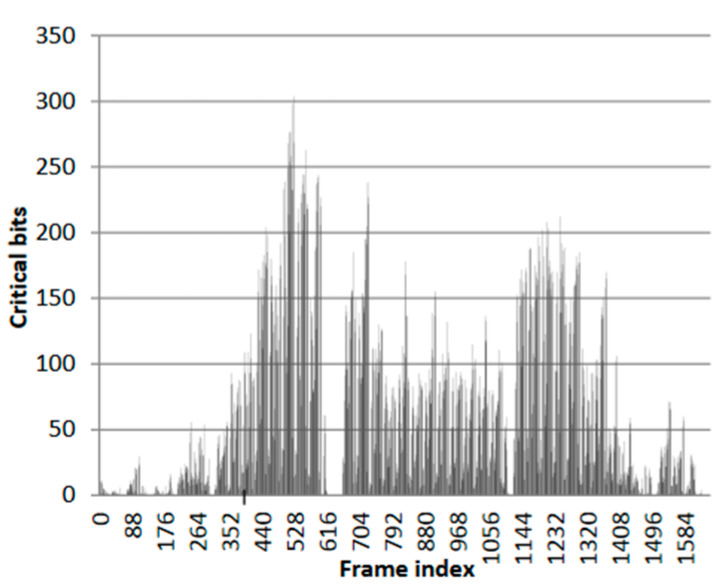
Number of critical bits in each frame of the misex3 test circuit [43].

**Figure 10 sensors-24-05356-f010:**
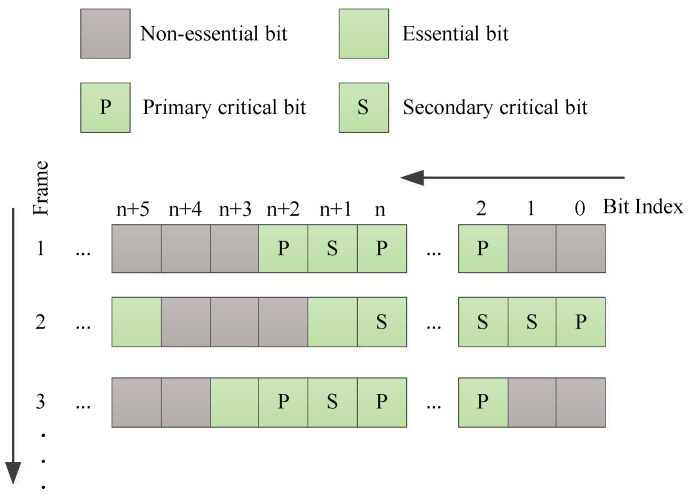
Distribution of non-critical bits, critical bits, primary critical bits, and secondary critical bits across different configuration frames.

**Figure 11 sensors-24-05356-f011:**
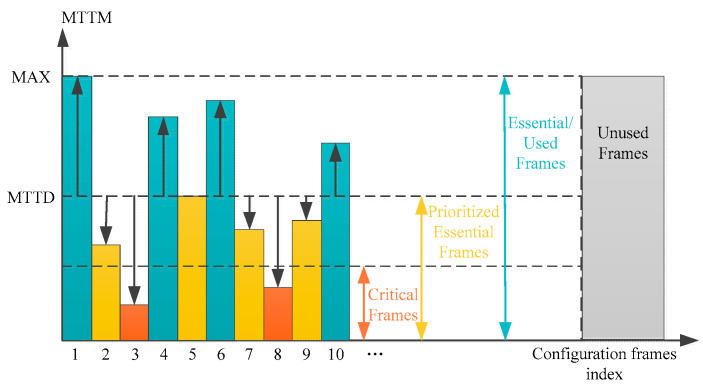
Classification of configuration frames in [71].

**Figure 12 sensors-24-05356-f012:**
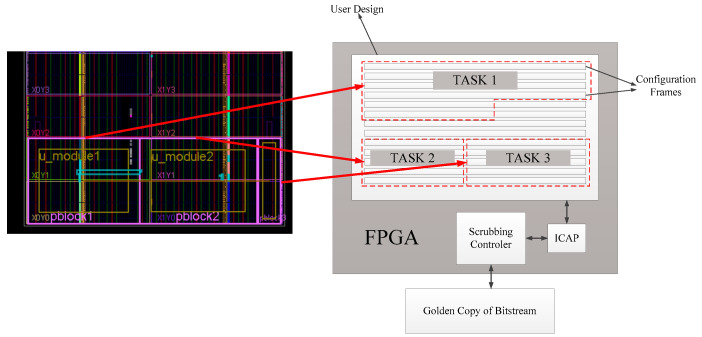
FPGA application-layer division [73].

**Figure 13 sensors-24-05356-f013:**
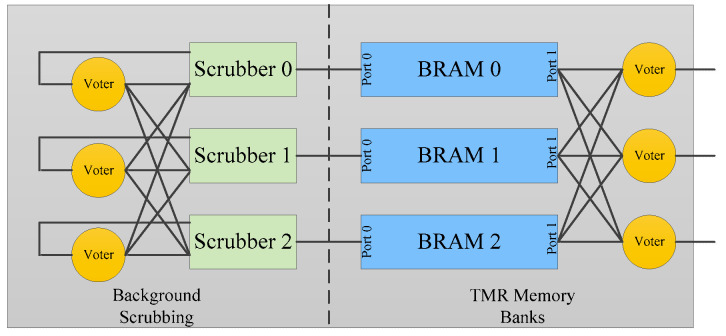
Memory scrubbing of BRAM [82].

**Table 1 sensors-24-05356-t001:** Summary of SRAM-based FPGA sensitivity analysis techniques.

Analysis Techniques			Advantages	Disadvantages
Analysis of application layer resources	Fault emulation-based	Using radiation tests	Emulate space radiation environments to the greatest extent.	High costs, time-consuming, potentially destructive to devices, and uncontrollable error locations.
		Using hardware emulation	High repeatability and minimal destructive impact on circuits.	Complex, specialized hardware is required to perform error emulation tasks.
	Analytical model-based		Has no damage to the circuit under analysis.	Long analysis time and low accuracy.
Analysis of configuration bitstream	Based on dual-layer mapping		A certain research foundation.	Limited research on newer FPGA models.
	Using fault injection		The principle is simple and easy to operate.	Performing a comprehensive error injection on a large-scale FPGA takes too much time.

**Table 2 sensors-24-05356-t002:** Summary of nonlinear configuration scrubbing methods.

Research	Description
[43]	Shifted scrubbing. Derive the optimal scrubbing starting frame address based on the number of critical bits in each frame.
[44]	Scatter scrubbing. Derive the optimal starting frame address and adopt a jumping read-back verification method.
[70]	Combine essential-frame scrubbing and full-device scrubbing.
[33]	Further subdivide critical bits into PCB and SCB.
[71]	Categorize configuration frames into unused frames, basic frames, priority basic frames, and critical frames.
[72]	Distinguish highly critical bits within the critical bits.

**Table 3 sensors-24-05356-t003:** Summary of multi-task scrubbing scheduling techniques.

Research	Scheduling Methods	Advantages	Disadvantages
[73]	Static scheduling method based on task criticality and execution time.	Implemented scrubbing scheduling in multi-circuit task scenarios.	Adopts a static scheduling algorithm, unable to adapt to dynamic hardware task lists.
[74]	Dynamic scrubbing scheduling method based on time windows.	Requires less storage space and can adapt to dynamic hardware task lists.	Reliability decreases in scenarios with frequent parallel circuit tasks. Discriminates against low-criticality tasks.
[75]	Heterogeneous application scrubbing scheduling method based on time windows.	Considers DSP-related and high-data-throughput application tasks.	Allocates very low scrubbing frequencies for tasks of low criticality.
[76]	Dynamic scheduling based on DVFS.	Reduces conflicts among scrubbing tasks, further enhancing reliability.	Allocates very low scrubbing frequencies for tasks of low criticality.
[77]	Negotiation-driven scrubbing scheduling.	Better manages conflicts between scrubbing tasks.	Requires substantial storage space when there are a large number of tasks.

## Data Availability

No data were used for the research described in the article.

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
