# Peer review of "A Review on Soft Error Correcting Techniques of Aerospace-Grade Static RAM-Based Field-Programmable Gate Arrays"

_sensors, 2024, doi:10.3390/s24165356_

Round 1
Reviewer 1 Report
Comments and Suggestions for Authors
This paper presents a review of soft error correction and fault mitigation techniques for SRAM-based FPGAs. The authors reviewed configuration memory scrubbing techniques, modular redundancy, fault injection, etc.
FPGA reliability and fault-tolerance has been a hot topic of research for decades as these devices are very important in space and military applications. Considering the amount of research on this topic, this paper is very brief and lacks essential information.
Future revisions of this paper could benefit from the following comments:
1) Include a discussion of FPGA applications in space showing the need for advanced fault mitigation techniques. Include data and studies showing the expected error rates of SRAM FPGAs in harsh environments.
2) Include a discussion of the already existing reviews and surveys addressing this topic clearly explaining what updated material your paper presents.
3) FPGA vendors provide solutions for reliable deployment of FPGAs in space applications. Include a discussion of these solutions. Below are few examples:
a. Rad-hard and space-grade FPGAs from AMD/Xilinx, Microchip, Lattice Semiconductor, etc.
b. FPGA design tools such as AMD/Xilinx Isolation Design Flow (IDF), Microchip Design Separation methodology, etc.
c. ECC support in FPGA block RAM
d. Configuration controllers, internal scrubber IPs, fault injection IPs, etc.
4) Several recovery techniques relays on dynamic partial reconfiguration feature of FPGAs (reconfiguration of an entire region of the FPGA to recover from multi-bit errors). This feature has not been discussed in the paper.
Comments on the Quality of English Language
Minor editing of English language required
Author Response
Thank you so much for your kind suggestions. We have modified our script accordingly. Please see the attachment.

Reviewer 2 Report
Comments and Suggestions for Authors
Dear Authors,
Thank you for this good review of the state of the art.
The discussion is clear, and the manuscript is organized sufficiently well. However some sentences should be improved in English grammar, there are some repetitions and sometimes the vocabulary is used incorrectly.
There is a critical issue about your review of the state of the art. In fact, you state that configuration scrubbers retrieve the correct configuration from a “Bitstream memory(Radiation-hardened)”. However, there are many interesting solutions to avoid the golden memory but still be able to correct multiple-bit upsets, even with high multiplicity. Some solutions are based on redundant configuration scrubbing, either by generating redundant frames in the CRAM or by mirroring frames between identical FPGAs of homologous boards. Other solutions exploit error correcting codes to detect and correct upsets in pre-defined clusters of frames. Please make your review broader by adding a section on techniques for redundant-configuration scrubbing and configuration self-repair in FPGAs, and cite the relevant works.
Please find below some specific minor suggestions
Key: L – line(s)
L26, based on -> depending on
L27, SRAM, antifuse, and Flash -> SRAM-, antifuse-, and flash-based
L31-33, you state that as the feature size decreases, the sensitivity to SEEs increases. This is incorrect. It is not true for non-planar technologies. Regarding SEEs in general (SELs for example), there are technologies at lower feature sizes more robust than older technologies (see CMOS 28nm vs 90nm for example).
L63, in many years of work in configuration scrubbing of FPGAs, I rarely heard the “bitstream scrubbing” term. The bitstream a file containing loading instructions and data to be loaded in the CRAM of the FPGA chip. Experts in the field mostly use the term “configuration scrubbing” or “CRAM scrubbing”.
L64, were sufficed – were sufficient
L103, SEEs -> SEUs
L106, normally, by “fault-injection” experts refer to the usage of the configuration access logic to alter CRAM bits. Radiation experiments are considered a method for accelerated testing, but not fault injection.
L228, with an knowledge of -> with a knowledge of
L290-292, The concept of configuration scrubbing was already defined in the Introduction, please avoid repetitions.
L474-475, You state there is a 1.5x additional delay. Please explain the source of this delay to the reader.
L495-497, Please check the font size of equations (1), (2), and (3). Define and use appropriate mathematic symbols instead of whole English words.
L630-632, please note that accessing BRAM configuration conflicts with the operation of logic in the fabric accessing the pertaining memory blocks. Care should be taken to resolve these conflicts for successful operation of the design and of the configuration scrubber.
L655, functioning -> operation
Comments on the Quality of English Language
Dear Authors,
Thank you for this good review of the state of the art.
The discussion is clear, and the manuscript is organized sufficiently well. However some sentences should be improved in English grammar, there are some repetitions and sometimes the vocabulary is used incorrectly.
There is a critical issue about your review of the state of the art. In fact, you state that configuration scrubbers retrieve the correct configuration from a “Bitstream memory(Radiation-hardened)”. However, there are many interesting solutions to avoid the golden memory but still be able to correct multiple-bit upsets, even with high multiplicity. Some solutions are based on redundant configuration scrubbing, either by generating redundant frames in the CRAM or by mirroring frames between identical FPGAs of homologous boards. Other solutions exploit error correcting codes to detect and correct upsets in pre-defined clusters of frames. Please make your review broader by adding a section on techniques for redundant-configuration scrubbing and configuration self-repair in FPGAs, and cite the relevant works.
Please find below some specific minor suggestions
Key: L – line(s)
L26, based on -> depending on
L27, SRAM, antifuse, and Flash -> SRAM-, antifuse-, and flash-based
L31-33, you state that as the feature size decreases, the sensitivity to SEEs increases. This is incorrect. It is not true for non-planar technologies. Regarding SEEs in general (SELs for example), there are technologies at lower feature sizes more robust than older technologies (see CMOS 28nm vs 90nm for example).
L63, in many years of work in configuration scrubbing of FPGAs, I rarely heard the “bitstream scrubbing” term. The bitstream a file containing loading instructions and data to be loaded in the CRAM of the FPGA chip. Experts in the field mostly use the term “configuration scrubbing” or “CRAM scrubbing”.
L64, were sufficed – were sufficient
L103, SEEs -> SEUs
L106, normally, by “fault-injection” experts refer to the usage of the configuration access logic to alter CRAM bits. Radiation experiments are considered a method for accelerated testing, but not fault injection.
L228, with an knowledge of -> with a knowledge of
L290-292, The concept of configuration scrubbing was already defined in the Introduction, please avoid repetitions.
L474-475, You state there is a 1.5x additional delay. Please explain the source of this delay to the reader.
L495-497, Please check the font size of equations (1), (2), and (3). Define and use appropriate mathematic symbols instead of whole English words.
L630-632, please note that accessing BRAM configuration conflicts with the operation of logic in the fabric accessing the pertaining memory blocks. Care should be taken to resolve these conflicts for successful operation of the design and of the configuration scrubber.
L655, functioning -> operation
Author Response

(The authors gave the same response as above.)

Reviewer 3 Report
Comments and Suggestions for Authors
The article presents an overview of methods for detecting and correcting memory errors occurring in FPGAs when operating in extreme conditions. The authors performed an extensive literature review and analyzed available solutions regarding soft error sensitivity and repairing techniques of aerospace-grade SRAM-based FPGA. In my opinion, the article deserves publication, but I have one comment: If we apply fault-injection through radiation, it affects both the application layer and the configuration layer. It is worth noting in the article that this makes soft-error analysis more difficult because errors induced in the configuration layer may cause further errors in the application layer.
Comments on the Quality of English Language
Please correct typos:
page 2, line 64: "were sufficed" should be "were sufficient"
page 9, line 324: pins’ should be pins
In many places the authors use the Saxon Genitive "FPGA's", I think that using "FPGA" or "… of FPGA" should be better.
Author Response

(The authors gave the same response as above.)

Round 2
Reviewer 1 Report
Comments and Suggestions for Authors
“We comprehensively review the soft error sensitivity analyzing and repairing techniques…”
sensitivity analyzing è Sensitivity analysis
“Also scrubbing techniques can benefit from the dynamic partial reconfiguration feature
of FPGAs, allowing for scrubbing the erroneous configuration frames during the circuits
are running.”
during the circuits are running è while the circuits are running
Include a clear definition of Dynamic Partial Reconfiguration (DPR). DPR design flow involves separating the design into static regions and reconfigurable regions. Partial bitstreams of the reconfigurable regions can be loaded into the FPGA at run-time.
Comments on the Quality of English LanguageMinor editing of English language required
Reviewer 2 Report
Comments and Suggestions for Authors
Dear Authors,
Thank you for your modifications.
However, there is still a critical issue about your review of the state of the art. You added references to some review works, but still an important branch of the relevant literature is missing from your introduction. Please try harder in improving your references, including solutions are based redundant-configuration, intermodular scrubbing, self-repair and error correcting codes.
Best regards
